# Efficacy of Facial Exercises in Facial Expression Categorization in Schizophrenia

**DOI:** 10.3390/brainsci11070825

**Published:** 2021-06-22

**Authors:** Francesco Pancotti, Sonia Mele, Vincenzo Callegari, Raffaella Bivi, Francesca Saracino, Laila Craighero

**Affiliations:** 1Department of Neuroscience and Rehabilitation, University of Ferrara, via Fossato di Mortara 19, 44121 Ferrara, Italy; pancottifrancesco@gmail.com; 2Department of Neuroscience, Biomedicine and Movement Sciences, University of Verona, 37134 Verona, Italy; sonia.mele@univr.it; 3Integrated Department of Mental Health and Addictive Behavior, Local Health Trust of Ferrara, 44124 Ferrara, Italy; v.callegari@ausl.fe.it (V.C.); r.bivi@me.com (R.B.); f.saracino@ausl.fe.it (F.S.)

**Keywords:** schizophrenia, physical training, emotion recognition, embodied cognition, sensorimotor system, facial expression, transitive actions

## Abstract

Embodied cognition theories suggest that observation of facial expression induces the same pattern of muscle activation, and that this contributes to emotion recognition. Consequently, the inability to form facial expressions would affect emotional understanding. Patients with schizophrenia show a reduced ability to express and perceive facial emotions. We assumed that a physical training specifically developed to mobilize facial muscles could improve the ability to perform facial movements, and, consequently, spontaneous mimicry and facial expression recognition. Twenty-four inpatient participants with schizophrenia were randomly assigned to the experimental and control group. At the beginning and at the end of the study, both groups were submitted to a facial expression categorization test and their data compared. The experimental group underwent a training period during which the lip muscles, and the muscles around the eyes were mobilized through the execution of transitive actions. Participants were trained three times a week for five weeks. Results showed a positive impact of the physical training in the recognition of others’ facial emotions, specifically for the responses of “fear”, the emotion for which the recognition deficit in the test is most severe. This evidence suggests that a specific deficit of the sensorimotor system may result in a specific cognitive deficit.

## 1. Introduction

Schizophrenia is characterized by heterogeneous positive and negative symptom constellations [1]. Negative symptoms refer to a diminution or absence of normal behaviors, they account for a large part of the long-term morbidity and poor functional outcome in patients with the disorder [2,3,4], and have been reported as among the most common first symptom of schizophrenia [5]. Although positive symptoms are generally effectively managed with available antipsychotic medications, limited treatment options are available for negative symptoms [6]. Negative symptoms can be categorized into two independent factors: diminished expression, and apathy [2]. Diminished expression mainly includes reduction in the expression of facial emotions. Interestingly, considerable research evidence indicates that facial mimicry contributes to accurate and efficient recognition of facial expressions [7,8] which is essential for social cognition [9]. Indeed, individuals with schizophrenia experience problems in face emotion recognition throughout the course of the disorder [10,11]. Rather than a general deficit that encompasses all emotions, schizophrenia may be associated with a more specific deficit in the processing of a subset of negative emotions, including sadness and fear [12,13,14,15,16]. Furthermore, these specific deficits have been demonstrated in individuals with first-episode psychosis [17], and in individuals who are “at risk” for psychosis [18], suggesting that they may serve as markers of risk [19].

The possibility that reduced expression of facial emotions contributes to the difficulty in recognizing facial expressions in patients with schizophrenia has recently been considered in a facial expression categorization test experiment. Results showed that patients’ identification rates were lower than healthy controls, specifically for the responses fear and sadness [20]. Interestingly, in healthy participants, the identification rates at the same test were significantly modulated by constraint facial postures [21]. Participants were required to label pictures randomly taken from four morphed continua between two emotional facial expressions, while they were submitted either to a lower or to an upper face posture manipulation. The stretching of the mouth (i.e., participants were required to maintain a stick horizontally between the teeth without touching it with the lips) increased the percentage of happy average responses, and the upper face posture manipulation (i.e., participants were asked to frown and then a Band-Aid was applied in proximity to corrugator muscle, covering the eyebrows and partially lowering the eyelids) increased the percentage of sad average responses. These effects were found only for intermediate values of the continuum, where the expressions were more ambiguous. The authors assumed that facial postural manipulation induced a specific sensorimotor activation which enhanced the detection of visual cues congruent with that sensorimotor state. Specifically, they concluded that the implementation of low-level movement details influenced the discrimination of ambiguous facial expressions differing for a specific involvement of those movement details [21]. This interpretation of the results is congruent with embodied cognition theories, which suggest that we understand others’ emotions by reproducing the perceived expression in our own facial musculature (facial mimicry), and the mere observation of a facial expression can evoke the corresponding emotion in the perceivers. Consequently, the inability to form facial expressions would affect the experience of emotional understanding [22,23,24,25,26].

Therefore, it is possible that, in patients with schizophrenia, an increase in the ability to perceive and perform facial movements leads to an improvement in the recognition of facial expressions. Indeed, facial expressions are principally the result of stereotyped movements of facial skin and fascia due to contraction of the facial muscles in certain combinations. Such contractions create folds, lines, and wrinkles in the skin and cause movement of facial landmarks such as mouth corners and eyebrows which represent the most salient aspects of most expressions [27,28]. Consequently, as happens for all skeletal muscles, specific facial muscle mobilization training could lead to an improvement in the ability to modulate the recruitment of motor units. Furthermore, literature reports that physical therapy approaches play an important role in the multidisciplinary treatment of people with schizophrenia [29,30,31,32,33].

Facial muscle rehabilitation is successfully applied in individuals with facial paralysis and distorted facial expressions and movements, secondary to a facial neuromotor disorder [34,35,36,37]. It is worth noting that the neuromotor control of facial movements appears different from the usual motor control mechanisms of skeletal muscles due to the limited ability of the facial muscles to provide feedback. Intrinsic muscle receptors and joint receptors, primary sources for peripheral proprioceptive feedback to the central nervous system, are few or absent in the face [38,39]. To overcome the physiological absence of proprioception which presumably renders the brain uninformed of facial muscle performance, the facial neuromuscular reeducation approach of neurological patients consists of movement exercises accompanied by surface EMG biofeedback or mirror feedback [37]. Individuals who are provided with precise, extrinsic feedback about facial muscle activity learn to recruit the appropriate motor units for the desired movement, as it happens in relearning movement patterns involving the peripheral neuromuscular system.

On this basis, the present work proposed a functional physical training of the facial muscles to patients with schizophrenia. To ensure that training was well received by patients and that it can be easily used as rehabilitation therapy in public health facilities, the use of extrinsic feedback has not been envisaged. To inform patients about the correctness of the execution of the proposed movements, each exercise was designed as a transitive action, which acts towards an object to achieve a certain goal (e.g., to move a light ball by blowing through a straw). As a result, the achievement of the goal performed the feedback function such as the EMG signal or the image in the mirror. The training protocol was based on the parameters individuated by the Facial Action Coding System (FACS) [28,40] which systematically assessed the muscular components of facial behavior. FACS identified the appearance changes related to facial movements and aimed to identify individual muscle contractions, focusing not on the expression of emotions, but on the production of spontaneous facial movements. FACS uses numbers to refer to the appearance changes associated with 33 facial muscle contractions, or action units (AUs). Most AUs refer to the contraction of single muscles, but some muscles always co-occur, or are capable of producing different movements. Thus, the correspondence between facial muscles and movements is not always direct. Therefore, the present training protocol was specifically designed to mobilize the face according to the AU profiles of happiness, fear, anger, and sadness. For example, the happiness profile is characterized by AU 6 + 12. The action descriptor of AU 6 is “cheek raiser”, while that of AU 12 is “lip corner puller”. Consequently, examples of the exercises proposed to enhance the ability to express happiness were “move the glasses placed on the nose upwards using the cheek muscles”, and “maintain a stick horizontally between the teeth without touching it with the lips”.

The main aim of the present study was to test the effects of facial muscle mobilization training in patients with schizophrenia on performance in a facial expression categorization test [21]. To this end, the patients were tested before and after the training period, and their results were compared with those of a control group of patients with schizophrenia housed in the same psychiatric residence who did not participate in the training. The time interval between the two test phases was the same in the two groups. We expected an improvement in the recognition of sadness and fear, expressions for which patients with schizophrenia show difficulty in recognition [20].

## 2. Materials and Methods

### 2.1. Participants

Twenty-four inpatient participants with schizophrenia-spectrum disorders (SSD) classified according to the International Statistical Classification of Diseases and Related Health Problems 10th Revision (ICD-10) participated in this study, gave their written informed consent, and were randomly assigned to the experimental (*n* = 12; 8 females; mean age = 53.3 years, standard deviation = 8.81) and control (*n* = 12; 3 females; mean age = 47.9 years, standard deviation = 9.7) group. For each participant, Table 1 reports ICD-10 codes and description of main symptoms.

All participants were native Italian speakers of Caucasian ethnicity (as were the models depicted in the stimuli) and reported having normal or corrected-to-normal visual acuity. The patients were hosted at the long-term psychiatric residence “Il Convento” and psychiatric day center “Il Convento” in Ferrara (Italy). Participants were unaware of the purposes of the study. The procedures were approved by the local ethics committee (ref.: EM255-2020_UniFe/170592_EM) and were in accordance with the guidelines of the Declaration of Helsinki. Both groups of participants underwent the facial expression categorization test [20,21] twice. During the month between the first (Test 1) and second (Test 2) test administration, the participants in the experimental group were involved in physical training, while the participants in the control group carried out the activities proposed by the nursing home.

### 2.2. Facial Physical Training

The participants of the experimental group were involved in the training protocol in groups of four, for three 45-minute sessions per week, for five subsequent weeks (total 15 sessions). The timing of the protocol is based on those of the closest protocol in the literature, namely the Action Observation Treatment (AOT; [41]). AOT is a rehabilitation approach that aims to reactivate the sensorimotor skills to perform transitive actions in neurological patients. Specifically, we adapted it as follows: We slightly lengthened the duration of each session (45 min instead of 30), reduced the number of weekly sessions (3 days a week instead of 5) and lengthened the number of weeks (5 instead of 4). The choice of making groups of 4 people was based on having a group that was either too big nor too small. The training was performed in a dedicated room in which there were chairs, tables and the disposable material necessary to perform the exercises. To help the experimenter, a staff member of the healthcare facility was also present. Each exercise was explained and shown by the experimenter to the whole group. Subsequently, each participant performed the exercise for about 5 min, alone or with the help of the operator. During each session an average of 5 exercises was proposed. Exercises varied each time, and each of them was proposed several times during the different sessions.

As anticipated in the Introduction section, the facial physical training protocol was designed to mobilize the face according to the AUs profiles of happiness, fear, anger, and sadness (Table 2) and the relative action descriptors (Table 3), reported by [40].

The considered action descriptors essentially concern the movements of the lips and the muscles around the eyes, and the facial exercises proposed mainly involved these two facial districts. As already stated, we did not want to use any kind of external feedback as is commonly the case with neurological patients [37]. Patients with schizophrenia do not willingly accept the application of electrodes, nor observe their image in the mirror. We therefore decided to invent a series of exercises that required the execution of transitive actions. In other words, actions aimed at some objects, the result of which (the maintenance or mobilization of the objects) performed the feedback function of the correct execution of the exercise. We have not been able to exploit the literature to use transitive actions for the facial muscles, as so far only transitive actions involving the upper or lower limbs have been described [41]. The list of exercises is reported in Table 4.

### 2.3. Facial Expression Categorization Test

#### 2.3.1. Procedure

The participants in the Experimental group were tested the week before and after the training period, which lasted five weeks. The participants in the control group were also tested twice and the time interval between the two test phases was the same as that of the experimental group.

Stimuli consisted of 88 pictures of two human face models (one female, one male), selected in a previous study [42], and used here with permission (Figure 1). They were the result of a morphed transformation of four original pictures of each model portraying different emotion expressions (anger, fear, happiness and sadness) [28]. This transformation gave rise to four continua, anger-fear, anger-sadness, fear-happiness and happiness-sadness, each composed of 11 images. The three figures in the center of the continuum were presented four times each, while the others were presented twice, as in previous studies [20,21]. The total number of stimuli presented was 224 (28 for each model in each continuum × 2 models × 4 continua), subdivided into four blocks of 56 trials each. For more technical details, see [20,21].

Participants sat in front of a 19-inch LCD monitor (resolution: 1.280 × 800 pixels; refresh frequency: 60 Hz) on which stimuli appeared on a grey background subtending an 11° × 17° square region around the fovea. Stimulus-presentation timing and randomization were controlled with E-prime V2.0 (Psychology Software Tools Inc., Pittsburgh, PA, USA).

Each trial began with the presentation of a central fixation cross lasting 250 ms, followed by the presentation in the center of the screen of a randomly selected stimulus, lasting 500 ms. After the stimulus disappeared, four text boxes, “anger” (*rabbia* in Italian), “fear” (*paura*), “happiness” (*felicità*), and “sadness” (*tristezza*), appeared on the screen in line. The order of the four text boxes was balanced between the participants. Participants were asked to say the name of the emotion the face most resembled, and the experimenter recorded the response on the data recording computer.

#### 2.3.2. Data Analysis

For each continuum, we grouped responses according to levels, as indicated in Figure 1. For each emotion, we calculated the sum of relative responses given in trials belonging to the less ambiguous level of the two continua in which the emotion was present. Specifically, for anger, we calculated the sum of responses given at level 1 of the anger-fear continuum and at level 1 of the anger-sadness continuum; for fear, the sum of responses given at level 3 of the anger-fear continuum and at level 1 of the fear-happiness continuum; for sadness, the sum of responses given at level 3 of the anger-sadness continuum and at level 3 of the happiness-sadness continuum; for happiness, the sum of responses given at level 1 of the happiness-sadness continuum and at the level 3 of the fear-happiness continuum. Therefore, for each emotion, we calculated the number of responses given in 48 trials. We decided to consider these trials since our group previously found [20] that patients with schizophrenia differed from controls in the number of responses given at the less ambiguous levels of this same test.

The number of responses for each emotion was entered into a three-way 2 × 2 × 4 repeated-measure analysis of variance (ANOVA) with group (experimental, control) as between-subject variable, and test order (1, 2) and emotion (happiness, fear, anger, sadness) as within-subject variables. All pairwise comparisons were performed using the Bonferroni post-hoc test. A significance threshold of *p* < 0.05 was set for all statistical analyses. Effect sizes were estimated using the partial eta square measure (*η_p_^2^*). The data are reported as the *mean* ± *standard error of the mean* (*sem*).

## 3. Results

The three-way ANOVA performed on the number of responses revealed that the group main effect (*F*_1,22_ = 4.129, *p* = 0.054, *η_p_^2^* = 0.158), the test order main effect (*F*_1,22_ = 1.676, *p* = 0.209, *η_p_^2^* = 0.070) and the two-way interaction test order × emotion (*F*_3,66_*=* 2.572, *p* = 0.061, *η_p_^2^ =* 0.105) were not significant. The emotion main effect was significant (*F*_3,66_ = 19.779, *p* < 0.001, *η_p_^2^* = 0.473). The post-hoc analysis indicated that the number of happiness (42.3 ± 1.11) and sadness (38.5 ± 1.13) responses were greater than the number of fear (32.5 ± 1.29; all *ps* < 0.001) and angry (32.8 ± 1.89; respectively *p* < 0.001 and *p* = 0.002) responses. Happiness and sadness did not differ from each other (*p* = 0.08), and nor did fear and angry (*p* = 1).

The two-way interaction test order × group (*F*_1,22_ = 22.765, *p* < 0.001, *η_p_^2^* = 0.508), the two-way interaction emotion × group (*F*_3,66_ = 5.515, *p* = 0.002, *η_p_^2^* = 0.200), and the three-way interaction test order × emotion × group (*F*_3,66_ = 3.567, *p* = 0.018, *η_p_^2^* = 0.139) were significant.

Post-hoc analysis of the two-way interaction test order × group showed that in Test 1, the number of responses of the experimental (36.7 ± 1.31) and control (35.7 ± 1.49) groups did not differ (*p* = 1.000), and in Test 2 they were significantly greater for the experimental group (experimental, 40.1 ± 0.84; control, 33.7 ± 1.58, *p* = 0.014). The number of responses increased significantly in Test 2 for the experimental group only (experimental, *p* = 0.002; control, *p* = 0.134).

The post-hoc analysis of the three-way interaction test order × emotion × group showed a significant increase in the number of responses fear of the experimental group between Test 1 (30.9 ± 2.45) and Test 2 (38.3 ± 1.27, *p* < 0.001). Within each emotion, no other differences between the responses of the same group to the two tests (all *ps* > 0.08), or between the groups within the same test (all *ps* > 0.213), were significant (Figure 2).

## 4. Discussion

The aim of the present study was to test the effects of facial muscle mobilization training in patients with schizophrenia on performance in a facial expression categorization test [21]. The hypothesis was that, in patients with schizophrenia, the reduction in the expression of facial emotions is among the possible causes of problems in the recognition of others’ facial emotions. To this end, the patients were tested before (Test 1) and after (Test 2) a training period during which the lip muscles and the muscles around the eyes were mobilized through the execution of transitive actions. The data obtained in the two sessions of the facial expression categorization test were compared with those of a control group of patients with schizophrenia who did not participate in the training. Results showed a positive impact of the physical training in the recognition of others’ facial emotions. Specifically, the number of responses increased significantly in Test 2 with respect to Test 1 for the experimental group only. Furthermore, in Test 1, the number of responses of the two groups did not differ, and in Test 2 they were significantly greater for the experimental group. Previous data showed that the identification rate at the same facial expression categorization task was statistically significantly lower in patients with schizophrenia than in matched healthy participants [20]. The difference in performance was constantly present during trials requiring to produce the response fear, and less dramatically during trials requiring the response sadness. These findings are in agreement with evidence in literature that indicates that schizophrenia may be associated with a specific deficit in the processing of a subset of negative emotions including sadness and fear [12,13,14,15,16]. Therefore, we expected an increase in the ability to recognize the emotions for which the deficit is present. The results showed a specific increase in the number of responses of “fear”, the emotion for which the recognition deficit on this test is most severe.

The hypothesis underlying the present study arises from the claims of the embodied theories of cognition [43,44,45,46], according to which, stimulus recognition is supported by the automatic re-instantiation of modality-specific states captured during perception, action, and interoception of past experience with the stimulus [22]. In particular, the ability to recognize and understand the facial expressions of others depends on the automatic evocation of the same facial expression in the observer (i.e., spontaneous mimicry; [47,48,49,50,51]). Indeed, evidence shows that preventing participants from engaging expression-relevant facial muscles can impair their ability to detect briefly presented or otherwise ambiguous facial expressions that involve that specific muscle [52,53,54,55,56,57]. Interestingly, a consistent finding in the literature on emotion in schizophrenia is that individuals with schizophrenia are less facially expressive than individuals without schizophrenia in response to a variety of contexts and evocative stimuli [58]. They display fewer facial expressions in response to emotionally evocative film clips [59,60,61,62,63,64,65,66], foods [59], and social interactions [60,66,67,68,69,70,71,72,73]. Importantly, diminished expression is observed among individuals with schizophrenia both on [59,74] and off medication [63,64,65]. Therefore, we assumed that, as happens for all skeletal muscles, a physical training specifically developed to mobilize facial muscles could improve the ability to perform facial movements, and, consequently, even spontaneous mimicry and the related facial expression recognition. The positive influence of the physical training in the facial expression categorization test used in the present study supports this possibility. Indeed, this test was demonstrated to be a sensitive method to prove the role of the sensorimotor system in the perception of others’ actions, given that in healthy participants a facial postural manipulation influenced the identification rates [21].

Facial expressions are extremely relevant to social cognition. Information on the others’ affective states (e.g., others’ emotions) and on the environment (e.g., dangers from fearful reactions) could be extracted from facial expressions [9]. Several are the interventions that attempt to ameliorate deficits in social cognition in clients with schizophrenia. Targeted interventions train clients on aspects of one specific domain of social cognition [75,76,77]. The “training of affect recognition” (TAR) [78,79] primarily targets impairments in facial affect recognition, and it may be considered the most representative one. It is based on errorless learning, over-learning, and immediate positive feedback and feature abstraction. Since intensive coaching and modeling by the therapist is necessary, working with groups of more than two patients does not appear to be feasible. Therefore, despite the evidence of the positive effects of the TAR [78,79,80], the cost of personnel for this intervention is high. The present study proposes a less expensive alternative readily accepted and performed by patients.

This exercise protocol could be included among the different exercise therapies that have been proposed as an adjunct treatment in the multidisciplinary care of people with schizophrenia. Numerous randomized controlled trials [81] and meta-analyses [82] have shown a positive effect of exercise in treating schizophrenia. Specifically, there is evidence that aerobic, strength exercises, and yoga reduce psychiatric symptoms, state anxiety, and psychological distress and improve health-related quality of life. Furthermore, aerobic exercise improves short term memory, and progressive muscle relaxation reduces state anxiety and psychological distress [32]. These data, therefore, affirm the importance of the role of fitness trainers in meeting the mental and physical health needs of people with mental illness [33]. However, mental health facilities often do not have gyms or large spaces available for aerobic or fitness physical activity, and, furthermore, in many countries, the fitness trainer is not part of the multidisciplinary team working in mental health settings. The present training protocol, however, is based on performing transitive actions in a non-aerobic context where users are sitting or standing. Transitive actions are meaningful gestures implying the use of an object. Some examples of the proposed exercises are to inflate balloons, to push a small ball towards a target by blowing through a straw, to hold round objects of various sizes by tightly contracting the muscles around the eyes. To train in this type of action, a specific education in fitness is not necessary. Instead, education as psychiatric rehabilitation therapist is required, which develops the skills and knowledge necessary to work with patients with psychiatric disabilities, covering topics that include skills training, cognitive rehabilitation, and motivational strategies. This figure is normally part of the group that operates in mental health contexts, with the role of putting into practice the rehabilitation and educational interventions proposed by the therapeutic project. Therefore, the rehabilitation intervention proposed here can be easily applied by mental health services without the need for new professional figures or spaces with specific characteristics.

There are several limitations of this study that require further investigation to verify the outcome of this treatment. The heterogeneity of the pathological characteristics of the small sample studied may have influenced the results. Further studies are also needed to verify the duration of the effects of this training over time, and possibly propose changes to the timing and proposed exercises. Finally, it is necessary to investigate whether this protocol positively influences not only the ability to recognize but also to express emotions, and whether its effects lead to an overall improvement in social cognition.

By concluding, the results of this study bring for the first time evidence in favor of the efficacy of a rehabilitation approach dedicated to the training of sensorimotor skills in improving cognitive functions of those suffering from a severe mental disorder. They also suggest a broader application of mental health interventions, i.e., they must not be limited to social and psychological approaches, but must also include approaches based on the body and its interaction with the environment. The demonstration that an intervention to increase the ability to perform facial actions influences the perception of facial expressions indicates that a specific deficit of the sensorimotor system may result in a specific cognitive deficit, supporting the hypothesis that the sensorimotor system plays a central role in cognitive functions.

## Figures and Tables

**Figure 1 brainsci-11-00825-f001:**
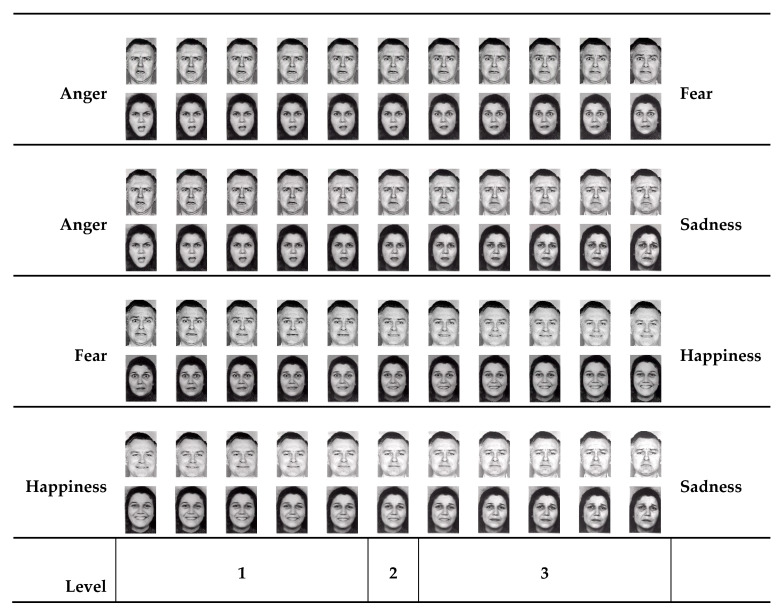
The 88 stimuli used in the facial expression categorization test, separated for each continuum.

**Figure 2 brainsci-11-00825-f002:**
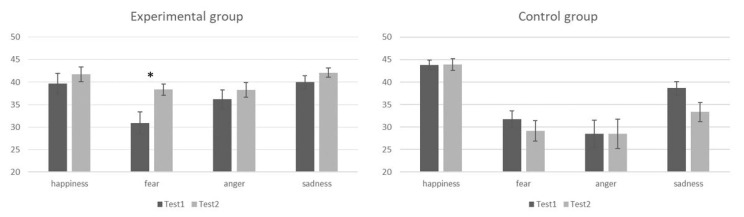
Mean values of responses for each emotion (happiness, fear, anger, sadness), for the experimental (leftmost panel) and control (rightmost panel) group, at Test 1 (black bars) and Test 2 (grey bars). * indicate statistically significant comparisons. Thin lines above histograms indicate standard error of the mean.

**Table 1 brainsci-11-00825-t001:** List of participants, indicating the relative age, gender, ICD-10 code, and the main symptoms. Specifically, the codes pertaining to the participants are: (i) F20.0 paranoid schizophrenia; (ii) F20.1 hebephrenic schizophrenia; (iii) F20.5 residual schizophrenia; (iv) F25.2 schizoaffective disorder, mixed type.

Experimental Group				Control Group			
Age	Gender	ICD-10 Code	Main Symptoms	Age	Gender	ICD-10 Code	Main Symptoms
57	M	F 20.5	Affective flattening; poverty of speech; reduced social functioning	33	M	F 20.0	Chronic persecutory delusional disorder; auditory hallucinations
62	M	F 25.2	Persecutory delusional ideas; emotional instability; relational isolation	50	M	F 20.0	Chronic persecutory delusional disorder
45	F	F 25.2	Affective instability; delusional persecutory cues; incongruity of thought	40	F	F 20.1	Alterations of affectivity; fluctuating hallucinations; disorganized behavior and thinking
57	F	F 20.0	Persecutory delusional ideation; auditory hallucinations	56	M	F 20.0	Chronic persecutory and erotomanic delusional disorder; auditory hallucinations
41	F	F 25.2	Affective instability; delusional persecutory cues; incongruity of thought	34	F	F 20.0	Chronic persecutory delusional disorder
35	F	F 20.1	Alterations of affectivity; fluctuating and fragmentary delusions; unpredictable behavior	54	F	F 20.0	Mystical Chronic Delusional Disorder
53	M	F 20.5	Affective flattening; poverty of speech; reduced social functioning	53	M	F 20.0	Chronic persecutory delusional disorder
63	F	F 20.1	Alterations of affectivity; fluctuating delusions and hallucinations; mannerisms; fatuous and inappropriate mood; disorganized thinking; incoherent speech	42	M	F 25.2	Persecutory delusional ideas; mind reading; emotional instability
55	F	F 20.1	Affective flattening; loss of initiative; social isolation	66	M	F 20.0	Chronic persecutory delusional disorder; apathy; abulia; relational isolation
52	M	F 20.5	Affective flattening; poverty of speech; reduced social functioning	56	M	F 20.0	Chronic persecutory delusional disorder; apathy; abulia; relational isolation
58	F	F 20.0	Chronic persecutory delusional disorder	49	M	F 20.5	Psychomotor slowdown; psychoaffective flattening; passivity and lack of initiative; poverty of speech
62	F	F 20.1	Affective flattening; loss of initiative; social isolation	45	M	F 20.5	Affective flattening; poverty of speech; reduced social functioning

**Table 2 brainsci-11-00825-t002:** List of expressions considered in the facial expression categorization test and relative FACS action units [40].

Emotion	Action Units (AUs)
Anger	4 + 5 + 7 + 23
Fear	1 + 2 + 4 + 5 + 7 + 20 + 26
Happiness	6 + 12
Sadness	1 + 4 + 15

**Table 3 brainsci-11-00825-t003:** List of FACS action units, relative action descriptors, and underlying facial muscles [40], of the expressions considered in the facial expression categorization test.

AU Number	Action Descriptor	Muscular Basis
1	Inner brow raiser	Frontalis (Pars Medialis)
2	Outer brow raiser	Frontalis (Pars Lateralis)
4	Brow lowerer	Depressor Glabellae Depressor Supercilii Corrugator Supercilii
5	Upper lid raiser	Levator Palpebrae Superioris Superior Tarsal Muscle
6	Cheek raiser	Orbicularis Oculi (Pars Orbitalis)
7	Lid tightener	Orbicularis Oculi (Pars Palpebralis)
12	Lip corner puller	Zygomaticus Major
15	Lip corner depressor	Depressor Anguli Oris
20	Lip stretcher	Risorius, Platysma
23	Lip tightener	Orbicularis Oris
26	Jaw drop	Masseter

**Table 4 brainsci-11-00825-t004:** List of facial exercises.

Facial District	Exercise
Mobilization of the lips	hold a stick horizontally between the teeth without touching it with the lips hold a stick horizontally between the lips without touching it with the teeth hold a stick vertically between the lips without touching it with the teeth hold objects of varying heaviness between the upper lip and nose hold round objects of various sizes and textures between the lips push a small ball towards a target by blowing through a straw make bubbles in different amounts of water by blowing through a straw inflate balloons with different resistance make soap bubbles
Mobilization of the muscles around the eyes	move the glasses placed on the nose upwards using the cheek muscles hold round objects of various sizes by tightly contracting the muscles around both eyes hold round objects of various sizes by tightly contracting the muscles around one eye, and look with the other eye remove a little piece of paper placed between the eyebrows by frowning

## Data Availability

The data presented in this study are openly available in FigShare at https://doi.org/10.6084/m9.figshare.14710638.v1. Dataset posted on 1 June 2021.

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
