# Peer review of "Efficacy of Facial Exercises in Facial Expression Categorization in Schizophrenia"

_brainsci, 2021, doi:10.3390/brainsci11070825_

Round 1
Reviewer 1 Report
The manuscript describes the effects of facial mobilization training on Schizophrenia patients. It evaluated whether the training improved their ability to recognize emotional expressions. Results indicated that the patients were able to recognize emotional facial expressions better than the baseline readings prior to the training especially for expressions of fear. Overall the paper is well organized and the introduction, related studies and experimental design is sound. The results and discussions are given adequate coverage and share valuable insights in the patients ability to recognize various facial expressions. However the facial physical training section needs some clarification. The reason behind the 5 weeks duration, 45 minutes duration, interval between the 3 sessions per week and groups of four needs clarification. It would be helpful to know if these durations and frequencies matter and would variations in them show different results. The paper should also provide some explanation on the choice of exercise activities. The activities were intended to mobilize the lips and muscles around eyes but were the activities chosen from an existing study or common knowledge? Overall the paper provides an interesting research topic and presents a method to potential improve ability to recognize facial expressions. The reviewer therefore recommends Accept after minor revision.
Reviewer 2 Report
It is an interesting paper about efficacy of facial exercises in facial expression categorization in schizophrenia. However, several revisions are required as follows:
- The authors have used the ICD-9 rather than ICD-10 as a diagnostic tool. The reason why the author have used the 1CO-9 should further explained. In addition, schizophrenia-spectrum disorders are not one of the diagnostic codes in the ICD-10. Thus, detailed diagnosis of schizophrenia-spectrum disorders should be described. Moreover, it is additionally described whether the psychiatrists have made a diagnosis of schizophrenia spectrum disorders or not.
- The limitations of study should be also described. Most of all, discussion about the small size of sample should be further described. In addition, the potential influence of the heterogeneity of the study subjects on the findings should be explained. Moreover, a direction and method for further study should be discussed.
Author Response
Please see the attachement

Reviewer 3 Report
This is carefully designed field investigation of the effect of facial physical training on the improvement of emotional recognition, measured by means of facial expression categorization tests. The cognitive-affective impairment underlying schizophrenia is manifetsed on the level of poor processing and expression of emotional stimuli. This study provides insight into the resources of relatively simle approach for improvement of the deficits presented in mimic expression. Given the small sample size further studies are needed in order to verify the proposed rehabilitation approach and outcome. Authors are advised to include limitations section in the manuscript.
Round 2
Reviewer 2 Report
It is well revised according to reviewers' comments.